# Comparison of Predictive Models with Balanced Classes Using the SMOTE Method for the Forecast of Student Dropout in Higher Education

**Vaneza Flores** [1,2]**, Stella Heras** [2] **and Vicente Julian** [2,*]

1 Professional School of Systems and Computer Engineering, National University of Moquegua (UNAM), Moquegua 18000, Peru; vfloresg@unam.edu.pe
2 Valencian Research Institute for Artificial Intelligence (VRAIN), Universitat Politècnica de València (UPV), 46022 Valencia, Spain; sheras@dsic.upv.es
* Correspondence: vjulian@upv.es

**Abstract:** Based on the premise that university student dropout is a social problem in the university ecosystem of any country, technological leverage is a way that allows us to build technological proposals to solve a poorly met need in university education systems. Under this scenario, the study presents and analyzes eight predictive models to forecast university dropout, based on data mining methods and techniques, using WEKA for its implementation, with a dataset of 4365 academic records of students from the National University of Moquegua (UNAM), Peru. The objective is to determine which model presents the best performance indicators to forecast and prevent student dropout. The study aims to propose and compare the accuracy of eight predictive models with balanced classes, using the SMOTE method for the generation of synthetic data. The results allow us to confirm that the predictive model based on Random Forest is the one that presents the highest accuracy and robustness. This study is of great interest to the educational community as it allows for predicting the possible dropout of a student from a university career and being able to take corrective actions both at a global and individual level. The results obtained are highly interesting for the university in which the study has been carried out, obtaining results that generally outperform the results obtained in related works.

**Keywords:** university student dropout; predictive model; data mining; SMOTE

## 1. Introduction

The phenomenon of university student dropout has been a research topic for several decades [1]. At present, this problem has become more important due to the negative effects caused by COVID-19. With the pandemic, university dropouts have risen, as reported by UNESCO, with around 23.4 million students dropping out of university in Latin America and the Caribbean regions [2]. However, there is no consensus on the definition of dropout in relation to university studies [3,4]. Following Tinto's theory [5], desertion is defined as the procedure carried out by a university student when he/she voluntarily or forcibly abandons their studies, due to a negative or positive influence of internal or external factors.

The deficient detection of the dropout of students in higher education is an important problem for the managers of the world's universities [6–8]. The situation is exacerbated in countries with low cultural and economic levels and, hence, how to counteract the consequences of university student dropout is studied worldwide [9,10]. Consequently, research has emerged in various areas such as psychology, economics, sociology, health, education, computer science, and others, where there is a significant number of students who drop out from these type of university degrees. Many studies establish that the critical point where student dropout is manifested is in the first year of university studies [11–13]. However, student dropout is a phenomenon that can manifest itself in any academic

period [14,15], for which corrective and preventive measures should be proposed aimed at university retention.

Dropout, as a social problem in the university ecosystem, has negative effects in the socio-economic, institutional, academic, and individual environment [2]. In this sense it is necessary to have adequate tools that allow for the detection of student dropout in higher education institutions. Therefore, taking technology as an ally for solving student dropout problems, and considering the excessive volume of data administered by computer systems in universities, our approach is framed in a technological solution based on educational data sciences (EDC) [16,17], especially educational data mining (EDM), a discipline widely studied by researchers in order to address the analysis of education and learning in university students, such as in [18–20].

In this work, several methods and EDM techniques are applied and evaluated for the construction of a predictive model that allows for predicting student dropout at the National University of Moquegua (UNAM), a university licensed by the Superintendency of National University Education (SUNEDU) in Peru. The model uses a dataset of 4365 student academic records, from 2008 to 2019, provided by the Directorate of Academic Affairs and Services (DASA) of UNAM, and applies classification techniques such as decision trees, decision rules and Bayesian networks, using WEKA as a data mining (DM) tool. The work carried out can be of great interest to the university on which the analysis was performed, as well as in general for the entire educational community. In this sense, the proposed study makes it possible to predict the potential dropout of a student from a university course and, in this way, to be able to take corrective actions both globally and individually. After the development of the experiments, the results obtained were highly interesting as they outperformed the results obtained in the analyzed related works.

There are other authors that have investigated the application of these techniques to detect dropout risks. Section 3 shows how our model outperforms these proposals. In addition, when these predictive models are built with real data from the university ecosystem, it has been observed that the classes are unbalanced. Although researchers recommend carrying out adequate balancing of the classes [21] and other studies demonstrate the effectiveness of such techniques in other contexts (e.g., school students [22–24]), few have followed this approach, especially for higher education domains [25]. This study implements this recommendation and applies the SMOTE (synthetic minority over-sampling technique) algorithm for class balancing with synthetic data [26,27].

The rest of the work is structured as follows: Section 2 reviews related work; Section 3 develops the methodology used for our study; Section 4 shows the results obtained; Section 5 discusses the results by comparing them with other studies; Section 6 summarizes the findings of the investigation; finally, Section 7 provides additional information such as funding, conflicts of interest, future work, and acknowledgments.

## 2. Related Work

The precedents that address the problem of university dropout are diverse. For this study, we considered those focused on proposing techniques, tools, algorithms, and attributes to build a predictive model with a higher performance. In [28], the authors proposed a predictive model based on the Random Forest algorithm for the early detection of university dropout, using the statistical program R. The sample consisted of 17,910 students and almost 3000 variables that corresponded to the factors associated with dropout, considering the student's background and the current academic year or semester. For the model validation process, they employed the 10-fold cross-validation technique. The results obtained showed an AUC metric of 0.86 representing the area under the ROC curve. Random Forest has also been used to predict student dropout in other contexts, such as in MOOCs (massive online open courses) students, with an accuracy of 87.5%, AUC of 94.5%, precision of 88%, recall of 87.5%, and F1-score of 87.5% [29].

In [30], a review of the techniques, tools, algorithms, and attributes for data mining used in student dropout was developed. This research filtered those articles with common

elements to detect dropout in the academic context. The search was carried out in digital libraries, indexed journals, and institutional repositories, among other scientific databases. The results reported J48 as the best algorithm to be used in predictive models, compared to Naive Bayes and Random Forest, and showed the highest score for the classification technique for data mining, followed by the techniques of regression, clustering, and association rules. Likewise, the best data mining tool placed WEKA in first place, followed by Orange and Rapid miner. In the case of attributes, these were classified into three dimensions, namely: academic attributes, demographic attributes, and family attributes.

In [31], a systematic review of the techniques and tools for the prediction of university dropout using educational data mining techniques was carried out. For this purpose, studies were collected to predict university student dropout in traditional courses. Out of 241 studies identified in Scopus and Web of Science (WoS) catalogues, 73 studies were selected that focussed their research on data mining techniques for predicting student dropout in universities. It was concluded that the most frequently used educational data mining technique was decision tree (67%), followed by Bayesian classification (49%), neural networks (40%), and logistic regression (34%). They also reported that the most frequently used software tools were WEKA, SPSS, and R.

In [32], academic success was predicted by building two predictive models using Random Forest. The first proposal was aimed at accurately predicting whether a student would complete their program. The second implemented a model to predict, considering the students who complete their program, which specialization they would complete. For the first model, a sample of 38,842 student records from the University of Toronto was used, and for the second, the results of the first model were taken from the records of students who did complete their program of study. In both cases, the records that did not correspond to academically successful students represented the group of university dropouts. The results reported an accuracy of 91.19% for identifying students who completed their program, and for the 418 students who did not complete their program the accuracy was 52.95%. The average result for the first case presented an accuracy of 78.84% over the test set.

In [21], the authors identified the factors associated with the phenomenon of university dropout using educational data mining, taking the National University of Colombia (UNAL) as a case study. A sample of 655 records was taken from 2009 to 2015. Educational data mining techniques were used using five classification algorithms: RandomTree, J48, REPTree, JRip, and OneR. WEKA software was used for the data analysis. The research proposed the use of 20 attributes. The results showed better values for the J48 algorithm, with correctly classified instances (ICC) of 95.43% (J48), 89.51% (Random Tree), and 85.4% (REPTree; the lowest), for the group of decision trees. For induction rules 95.12% (JRip) and 87.05% (OneR) were obtained.

In [25], data mining techniques were used to identify and expose the importance of the variables that motivated university students to drop out of their studies at the Universidad Católica del Norte (UCN) in Antofagasta and Coquimbo (Chile). A sample of 9195 individuals representing the student community of 12 engineering careers at UCN was taken from 2000 to 2013. The study used data mining techniques to define the pattern of students. Three classifiers were constructed to identify students who dropped out and did not drop out of university studies based on Bayesian networks, neural networks, and decision trees, analysed with WEKA. The data analysis considered 14 attributes to characterize the dropout student, parameterized in the academic aspect. The results reported correctly classified instances (ICC) with 80%, 82%, and 76%, respectively, for each classifier. With respect to the ROC curve, the values were 83%, 74%, and 76%, respectively, showing the predictive model built with the decision tree algorithms as the best classifier.

In [33], the authors built a model based on data mining techniques to analyze the problem of student dropout at the Universidad Arturo Prat, Chile. The input data were classified into two categories: socioeconomic and academic. Methods implemented in WEKA were used to select variables. Subsequently, the classification algorithms of the categories, decision trees, Bayesian methods and neural networks, were analyzed using

cross-validation with 10 folders for the training process. The results show Random Forest as the best classification algorithm with 89.87% accuracy, beating the values of ZeroR with 73.03%, J48 with 89.49%, SimpleCART with 89.09%, NaiveBayes with 87.29%, BayesNet with 87.72%, and Multilayer Perceptron with 87.18%. In addition, Random Forest presented a value of 0.913 for the area under the ROC curve (AUC), exceeding the values of 0.499, 0.8556, 0.878, 0.902, 0.902, and 0.868 for each model, respectively.

In [12], the authors identified influential factors in the phenomenon of student desertion in the community of the computer engineering course at the Gastón Dachar University (Argentina) using data mining techniques. A sample of 855 cases of students was considered in the period from 2000 to 2009. The research used the classification technique in data mining and the WEKA tool to report which were the characteristics and the associated factors that had a higher incidence in suspended students or dropouts. Three classification algorithms, J48 (C4.5), BayesNet (TAN), and OneR, were proposed to assess the degree of accuracy. The results reported similar accuracy percentages for the three classification algorithms and differed in the determination of attributes, with some attributes being more significant than others in each algorithm. The reported values were assigned an ICC of 80.234% for J48, 78.129% for BayesNet, and 76.608% for OneR, and for accuracy, which defines defectors, a value of 79.7% was obtained for J48, 75.3% for BayesNet, and 72.1% for OneR.

As shown in this section, in recent years, there have been several works that have tried to provide a possible solution to the problem of university dropout. Although we focused our analysis on work that applied machine learning and EDM techniques to predict student dropout, other techniques have been investigated for the detection and adoption of strategies for preventing and reducing student dropout. For instance, evaluating the cognitive functions and learning skills of students [34], using non-linear panel data models to analyze the determinants of student dropout [35], or applying learning analytics to identify students at risk of dropout [36].

However, the analysis carried out shows the need to apply a methodological approach that allows for an adequate analysis of the factors involved in this problem. On the other hand, a clear imbalance in the data can be corrected using the appropriate technique. The predictive models generated in the analyzed related works showed acceptable results in the metrics used; however, only a class balancing using the SMOTE technique was developed in a few cases. Our study also implemented a predictive model optimizing the results with SMOTE, but, additionally, a comparative scheme of three typologies of data mining techniques was established. Finally, real data covering several graduations of students are necessary in order to have a really useful tool. In the related works, it was found that the studies used smaller cohorts in most cases, which sometimes did not include an entire academic cycle. The following sections describe our proposal, which attempts to address the weaknesses detected in these previous works.

## 3. Methodology

The aim of this study is to analyze predictive models and to determine the best model that reports the best performance indicators for the prognosis of university student dropout. The actions are focused on proposing and comparing the accuracy of eight predictive models built with classification techniques using balanced classes, for which the SMOTE method is used.

### 3.1. SMOTE

The method is based on the SMOTE (synthetic minority over-sampling technique) algorithm, which consists of rebalancing the dataset for the original training, combining the over-sampling and under-sampling of the different classes that form the dataset. The dynamics of the SMOTE algorithm are the creation of synthetic data based on the interpolation of the instances contained in the minority class [26,27]. The new data are created by taking the distance between individuals in the same neighborhood, and the generated

distance is multiplied by a value between 0 and 1 randomly, and depending on the number of synthetic data that are determined to increase for each observed data of the seed, a minority class is obtained, resulting in the different balanced classes [37,38]. Figure 1 shows the representation of the synthetic data generation using SMOTE.

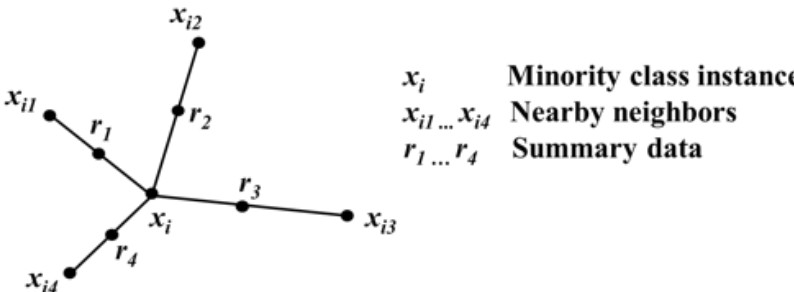

**Figure 1.** Representation of synthetic data points in the SMOTE algorithm.

## 3.2. CRISP-DM Methodology

In this work, the CRISP-DM methodology (cross industry standard process for data mining), widely used for data mining projects, was used to process and analyze the data [39]. CRISP-DM is a hierarchical process model and presents six phases structured in generic tasks, mapped into specific tasks with process instances. Figure 2 shows the process diagram, based on the CRISP-DM methodology, for this case study.

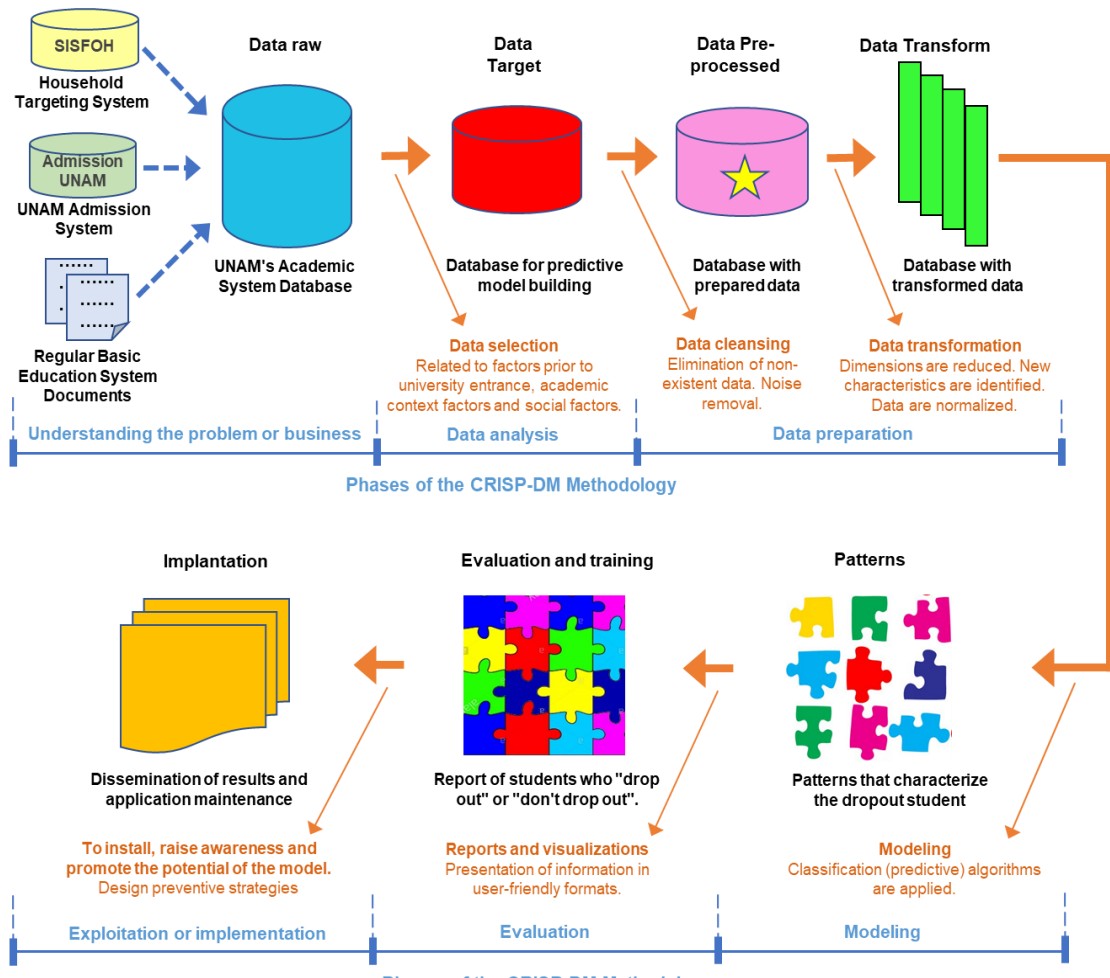

**Figure 2.** Process scheme aligned to the CRISP-DM methodology.

The phases of the CRISP-DM methodology are developed in the following sections:

Phase 1: Understanding the Problem or Business

The objectives and requirements of the data mining project are set out from the business (educational service) approach. This phase integrates the tasks necessary to identify the business objectives, assess the initial situation, determine the data mining objectives, and make the project plan.

UNAM, as a case study, presents five institutional objectives set out in the Educational Model 2020 [40]: (OB1) strengthen institutional management; (OB2) improve cultural extension and social outreach activities for the community; (OB3) improve academic training for university students; (OB4) promote formative, scientific, techno-logical, and humanistic research in the university community; and (OB5) implement disaster risk management. It can be seen that OB3 is aligned with the problems identified in the university educational service, understanding that the increase in the student dropout rate is closely related to the academic training given to UNAM students. Likewise, identifying the causes (behavioral patterns) and effects (negative scenarios) of student dropout could provide relevant information to the university authorities to rethink actions or generate new initiatives to prevent student dropout, and, in this way, reduce the student dropout rate at UNAM.

Since 2008, UNAM has had an academic system that stores personal and academic student data. However, it lacks socio-economic information, as well as the average grade of secondary education studies and the university entrance grade.

The data mining objectives set out for this study helped us to determine the scope of the predictive model. For this purpose, one general objective (GO) and four specific objectives (SO) were constructed. GO: To propose a predictive model based on artificial intelligence techniques that allows, from the input data, for detecting student desertion at UNAM, as follows: (SO1) determine the factors prior to university entrance that explain student desertion at UNAM, (SO2) determine the factors of the student university academic context that explain student dropout at UNAM, (SO3) determine the student social factors that explain student dropout at UNAM, and (SO4) train a predictive model based on artificial intelligence to detect student dropout at UNAM.

This phase ends with the formulation of the data mining project plan based on CRISP-DM, with the six phases and their respective general and specific tasks.

Phase 2: Data Analysis

In this phase, a process of analytical work is carried out. The objective is to begin the exploration of the UNAM academic system data, identifying the attributes required for the construction of the dataset, organized into three categories of factors prior to university entrance, factors of the university academic context, and the social factors of the students, comprised between the academic period of 2008-1 and 2019-1. This phase integrates the tasks necessary to collect the initial data (provided by DASA, as shown in Figure 3), data exploration, and data quality verification. In the latter, the consistency of the values present in each field has been verified. If a field has a null value (empty, not specified), the cells have been formatted and left empty. If data are ambiguous, they have been removed as inconsistent for further analysis. For the attribute year of birth and other missing personal data of the student, the Sistema de Focalización de Hogares (SISFOH) has been used. Finally, the dataset is obtained and is ready to be used in the data preparation phase.

Phase 3: Data Preparation

In this phase, the dataset provided in the previous phase is used and the selection of data relevant to the identified problem is performed. For this purpose, the WEKA tool is used, which allows for the activities of data cleaning, elimination of incomplete records that do not represent noise to the model, actions to complete missing data, and, finally, the construction of the target class, which in our case is called student status. Therefore, the tasks of phase 3 integrate the selection, cleaning, construction, integration, and formatting of data.

The data entered in text format are changed to numerical format and, in parallel, a data dictionary is designed with the labels for each attribute classification (entry mode, career, marital status, and other). For the student's date of birth attribute, with an absence of approximately 30%, 20% is completed with data from the Directorate of University Welfare and SISFOH. Some incomplete records are eliminated when verified that their percentage of presence in the dataset is minimal, approximately 4%. For the balancing of the target class student status, information is completed with synthetic data using SMOTE.

In the data transformation, the dimensions of the instances (records) and attributes of the dataset are reduced. New characteristics are also identified as being possibly important for the predictive model, such as the years spent at university and the years elapsed from the completion of secondary school (school leaving) to university entrance. Finally, the data type is standardized by setting the data to a numerical format for all values in the dataset.

The development of the phase 3 tasks resulted in the data structure shown in Table 1.

| Requested attributes | Remark | Requested attributes | Remark |
|---|---|---|---|
| 1. Registration number | Number generated to reserve the student's identity. | 14. Abigeo department | Department where the student was born. |
| 2. Sex | Student's gender. | 15. Abigeo province | Province where the student was born. |
| 3. Date of birth | Student's date of birth. | 16. Abigeo district | District where the student was born. |
| 4. Marital status | Student's marital status. | 17. College | Educational institution where the student completed high school. |
| 5. Professional career | Program of study chosen by the student. | 18. Abigeo of the school | District where the school is located. |
| 6. Faculty | Area where the student's program of study operates. | 19. Type of school | The type can be public or private. |
| 7. Curriculum | Curricular plan of the program of study. | 20. Year of school completion | Year in which he/she finished high school. |
| 8. Student class | Identifies dropouts or students who continue with their university studies. | 21. Branch code of entry | Code that indicates the branch where the student enters the university. |
| 9. Curriculum change | Only the year in which the change of enrollment took place appears. | 22. Description of entry branch code | Describes the branch (Moquegua, Ilo and Ichuña). |
| 10. Current branch of study | Current branch where the student is studying. | 23. Semester of entry | Semester of university entrance. |
| 11. Current branch of study | Branch in which the student entered. | 24. Semester of last enrollment | Semester of last registered enrollment. |
| 12. Mode of entry | Type of admission to the university through the ordinary or extraordinary admission process. | 25. Semester of graduation | Semester of university graduation. |
| 13. Abigeo country | Country where the student was born. | 26. Last cumulative average | Cumulative average grade. |

**Figure 3.** Attributes provided by DASA.

Phase 4: Modelling

Data mining techniques are used for the modelling process. Specifically, the algorithms of decision trees, decision rules, and Bayesian networks are used in order to carry out an adequate predictive analysis and to choose the algorithm with the best accuracy for detecting student dropout at UNAM. For the construction of the eight models and the development of the different data mining tasks, WEKA software is used. In this phase, the modelling technique is chosen, a test plan is generated, the models are built, and finally the evaluation process is carried out.

The input parameters for the selected algorithms run in WEKA use a total of 4756 instances, 18 attributes, and a cross validation with 10 folds. The specific information provided for the execution of each algorithm is detailed in Table 2.

Phase 5: Evaluation of Data Mining Techniques

In this next-to-last phase of the CRISP-DM model, reports and visualizations of the generated models are developed in order to establish a comparison of the results and to perform statistical analyses to determine the best predictive model. In our case, the Random Forest algorithm is the algorithm that reported the best accuracy and performance values, as shown in the following section.

**Table 1.** Attributes selected and grouped into the factors associated with university student dropout.

| Factor Associated with Student Desertion | | Name of the Selected Attribute | Data Type |
|---|---|---|---|
| Factors before university entrance | 1 | Type_School | Numeric |
| | 2 | Age_Entrance_School | Numeric |
| | 3 | Type_Entrance_U | Numeric |
| | 4 | Faculty | Numeric |
| | 5 | YearsBetween_University_Entry | Numeric |
| Academic context factors | 6 | YearIngress_U | Numeric |
| | 7 | SemesterIngress_U | Numeric |
| | 8 | Entry_Subsidiary | Numeric |
| | 9 | Current_Subsidiary_Studies | Numeric |
| | 10 | Professional_Career | Numeric |
| | 11 | Curricular_Plan | Numeric |
| | 12 | Curricular_Change | Numeric |
| | 13 | Last_Average_Accumulated | Numeric |
| | 14 | Semesters_of_Tenure_U | Numeric |
| | 15 | New_Application | Numeric |
| Social factors | 16 | Gender | Numeric |
| | 17 | AgeIncome_U | Numeric |
| **Target class** | | | |
| | 18 | StatusStudent | [dropout, not dropout] |

**Table 2.** Initial configuration values.

| Algorithms | Parameters |
|---|---|
| Random Forest | I = 100 iterations of trees in the random forest. K = 0 number of randomly chosen attributes. M = 1.0 minimum number of instances per leaf. V = 0.001 minimum proportion of variance present at a node for splitting to be performed on the regression trees. |
| Random Tree | K = 0 number of randomly chosen attributes M = 1.0 minimum number of instances per leaf. V = 0.001 minimum proportion of variance present at a node for splitting to be performed on the regression trees. |
| J48 | C = 0.25 confidence factor used for pruning M = 2 minimum number of instances per leaf. |
| REPTree | M 2 minimum number of instances per leaf V 0.001 minimum proportion of variance present at a node for splitting to be performed on the regression trees N = 3 data used for pruning L = −1 No restriction for maximum tree depth I = 0.0 Count of the initial value of the class |
| JRIP | F = 3 data used for pruning N = 2.0 minimum total weight of instances in one rule O = 2 optimization runs |
| OneR | B = 6 minimum size of the cube used to discretize the numerical attributes |
| Bayes Net | K = 2 number of randomly chosen attributes P = 1 maximum number of parents a node can have in the Bayes network A = 0.5 alpha, used to estimate the conditional probability tables |

Phase 6: Exploitation or Implementation

Finally, in this phase, the results of the predictive model are analyzed, which is the predictive knowledge of student desertion at UNAM, which must be transformed into

actions aligned with the business process (university education service) and be oriented to the objective of the student desertion problem. Thus, with the knowledge obtained, it is necessary to propose strategies to increase the retention rates of students at UNAM. A data mining project does not usually end with the presentation of the predictive model to the company or institution involved. The documentary aspect (implementation plans) and the training of the different types of users (teacher, career director, dean, academic vice-president, among others) are necessary. Therefore, it is important to complement this with actions that allow for the strengthening of university academic management to enable university student retention, in which strategies, regulations, and internal educational policies are proposed to reduce the university student dropout rate at UNAM. Likewise, the maintenance of the application (predictive model) and the people responsible for the dissemination of the results obtained should be considered in this phase.

## 4. Results

For this study, one general hypothesis and four specific hypotheses were proposed, which are aligned to the objectives of the data mining project.

### 4.1. Hypotheses

#### 4.1.1. General Hypothesis (GH)

The predictive model based on artificial intelligence techniques, from the input data, significantly predicts student dropout at the National University of Moquegua (UNAM).

#### 4.1.2. Specific Hypotheses (SH)

- SH1: Factors prior to the student's university entrance have a significant impact on student dropout at UNAM.
- SH2: The factors of the student's university academic context have a significant impact on student dropout at UNAM.
- SH3: The student's social factors have a significant impact on student dropout at UNAM.
- SH4: The training of a predictive model based on artificial intelligence allows, from the input of factors that characterize the pattern of the student deserting, for significantly detecting student dropout at UNAM.

### 4.2. Testing the Hypotheses

The following parameters and evaluation algorithms were used to test the hypotheses:

- Precision, defined as the representation of the number of true positives (TP) whose values are actually positive compared to the total number of true positives and false positives (TP + FP) predicted (1).

$$\text{Precision} = \frac{TP}{TP + FP} \tag{1}$$

- Area under the ROC curve (AUC), one of the main techniques for evaluating the performance of classification models, and widely used to measure the quality of probabilistic classifiers.
- Percentage of instances correctly classified (ICC), a measure used to interpret and evaluate classification algorithms in data mining projects, representing the value or percentage of instances that were correctly classified in the predictive model.
- Attribute selection algorithms and search method, the predictive model is built with a classification algorithm and 18 input attributes. SH1-3 were evaluated with five attribute selection algorithms and one search method.

As a result of the previous steps, the dataset for data analysis was obtained. The result was a subset of 18 attributes selected for the development of the predictive model, as shown in Table 3.

**Table 3.** Attributes selected and grouped in the factors associated with student dropout.

| Factor Associated with Student Dropout | Name of the Selected Attribute |
|---|---|
| Factors prior to university admission | 1. School type |
| | 2. College graduation age |
| | 3. Entry Mode |
| | 4. Faculty |
| | 5. Years Between Leaving College to Entering University |
| Factors of the academic context | 6. Year of admission |
| | 7. Semester admission |
| | 8. Campus admission |
| | 9. Current Campus admission |
| | 10. Degree |
| | 11. Curriculum Plan |
| | 12. Curriculum Change |
| | 13. Last Accumulated Average |
| | 14. Semesters of Permanence |
| | 15. New application |
| Social factors | 16. Gender |
| | 17. Age of admission |
| Target class | 18. Status of the Student |

Analyzing the performance measures of the eight predictive models with the parameters accuracy, area under the ROC curve (AUC), and the percentage of correctly classified instances (ICC), Table 4 shows that the best accuracy result was obtained by the Random Forest algorithm with a value of 0.97, which indicates that the predictive model performed a correct classification of 4604 records out of a total of 4756 instances.

**Table 4.** DM technique evaluation.

| Algorithms | CCI (%) | ROC Area | Dropout | TP | FP | Precision | Accuracy |
|---|---|---|---|---|---|---|---|
| Random Forest | 96.78 | 0.99 | Not dropout | 0.96 | 0.03 | 0.97 | 0.97 |
| | | | Dropout | 0.97 | 0.04 | 0.96 | |
| Random Tree | 93.06 | 0.95 | Not dropout | 0.92 | 0.05 | 0.95 | 0.93 |
| | | | Dropout | 0.95 | 0.08 | 0.91 | |
| J48 | 95.63 | 0.98 | Not dropout | 0.95 | 0.03 | 0.96 | 0.96 |
| | | | Dropout | 0.96 | 0.05 | 0.95 | |
| REPTree | 95.40 | 0.98 | Not dropout | 0.95 | 0.04 | 0.96 | 0.95 |
| | | | Dropout | 0.96 | 0.05 | 0.95 | |
| JRIP | 94.74 | 0.96 | Not dropout | 0.94 | 0.041 | 0.96 | 0.95 |
| | | | Dropout | 0.96 | 0.064 | 0.94 | |
| OneR | 79.00 | 0.79 | Not dropout | 0.74 | 0.160 | 0.82 | 0.79 |
| | | | Dropout | 0.84 | 0.260 | 0.76 | |
| Bayes Net | 89.40 | 0.96 | Not dropout | 0.92 | 0.137 | 0.87 | 0.89 |
| | | | Dropout | 0.86 | 0.075 | 0.92 | |
| Naive Bayes | 89.47 | 0.96 | Not dropout | 0.92 | 0.135 | 0.87 | 0.89 |
| | | | Dropout | 0.86 | 0.075 | 0.92 | |

Regarding the ROC area, Random Forest obtained 0.993, the best value for the ROC area, in contrast with OneR with a value of 0.79. The scientific literature indicates that the best value is the one with a tendency to 1. This result allowed for qualifying the predictive model as a predictive proposal with a greater capacity to discriminate against students with a dropout prole. Likewise, it supports the quality of the entry values.

Finally, for the percentage of correctly classified instances (ICC), Random Forest obtained a value of 96.78%, in contrast with OneR with a value of 79.00%. This result is interpreted as the ability or robustness of the Random Forest algorithm to correctly classify the instances in the dataset.

For the selection of best attributes, attribute selection algorithms implemented in WEKA were used and for the evaluation of the attribute quality to discriminate the target class, the Ranker attribute search method was used. These algorithms remove those attributes that are irrelevant or have little influence on the target class (output variable) of the predictive model. It was concluded that all attributes from Table 3 had a significant impact on student dropout from UNAM, and were 100% relevant to predict the state (dropout/not dropout) of the target class (student state) of the predictive model.

Finally, Table 5 compares the performance of our models with those trained in other previous related work (results as published by the authors), illustrating the improvement achieved with the application of our SMOTE balancing technique.

**Table 5.** Comparison of our model with other related work.

| Authors | Evaluated Techniques | Best Technique | Balance Data (Y/N) | Dataset Size | Accuracy | ROC Area | CCI (%) |
|---|---|---|---|---|---|---|---|
| Flores, Heras and Julián (our model) | Random Forest, Random Tree, J48, REP Tree, JRIP, OneR, Bayes Net, Naive Bayes | Random Forest | Y | 4365 | 0.97 | 0.99 | 96.78 |
| Behr et al. (2020) | Random Forest | Random Forest | N | 17,910 | - | 0.86 | - |
| Beaulac and Rosenthal (2019) | Random Forest | Random Forest | N | 38,842 | 0.79 | - | - |
| Solis et al. (2018) | Random Forest, Neural networks, SVMs, Logistic regression | Random Forest | N | 80,527 | - | - | 91.00 |
| Hernández-Leal et al. (2018) | Random Tree, J48, REP Tree, JRIP, OneR | J48 | N | 655 | - | - | 95.43 |
| Maya et al. (2017) | Multilayer Perceptron, Random Forest, J48, Random Tree | Random Forest | N | 670 | 0.88 | - | 85.50 |
| Miranda y Guzmán (2017) | Neural networks, Decision Trees, Bayesian Nets | Decision Trees | Y | 9195 | - | 0.74 | 82.00 |
| Torres et al. (2016) | Random Forest, ZeroR, J48, Simple CART, Naïve Bayes, Bayes Net, Multilayer Perceptron | Random Forest | N | 5547 | 0.89 | 0.91 | - |
| Eckert and Suénaga (2015) | J48, Bayes Net (TAN), OneR | J48 | N | 855 | 0.79 | - | 80.23 |

## 5. Discussion

### 5.1. Testing Hypothesis 1

The study validates five attributes of the dataset, obtained by selecting the factors prior to university entrance through five selection algorithms implemented with WEKA. The selected attributes are Modality_Entry_U (university entrance modality), YearsBe-

tween_University_Entry (years elapsed from high school completion to university entrance), faculty, School_Type (public or private), and School_Entry_Age. The focus on the factor associated with previous university entrance coincides with the study by [28], who considered a broader detail of the characteristics prior to university entrance. In the present study, we only considered five of these attributes, because the scope was limited to the data provided by the DASA of UNAM and because we considered the theoretical model of [41] as a reference, which, in the pre-university characteristics, presented similar attributes to those studied in this research, coinciding with [28] in only two attributes: School_Type and YearsBetween_School_Leaving (attribute generated with the age of leaving secondary school).

Overall, the authors referenced in Section 2 support the significant impact of factors prior to university entry on student dropout. Furthermore, the performance of the model is validated by the area under the ROC curve parameter, which reported a value of 0.723. However, in [28], the area under the ROC curve had a value of 0.86. The difference could be supported by the pooled analysis of the entry attributes, noting that this study made an assessment of all of the factors associated with dropout, and in our case, the analysis was performed exclusively on the factors prior to university entry. However, both results reported a significant incidence of the attributes in the model.

### 5.2. Testing Hypothesis 2

The study validates ten attributes of the dataset, obtained by selecting the academic context factors through five selection algorithms implemented with WEKA. The selected attributes are Semesters_of_Tenure_U (period of permanence in the university), Last_Average_Accumulated, Curricular_Plan, Year_Ingress_U (year of admission), Professional_Career, Current_Subsidiary_Studies (branch), Subsidiary_Ingress (branch or branch of the career or programme where enrolled), SemesterIngress_U, Change_Curricula, and New_Application_U (new application in the same university). The consideration of the academic context factor in comparative studies and the evaluation of predictive models coincide with [8,25,33,42,43]. Research supports the significant impact of academic context factors on student dropout. Likewise, the ability to correctly classify the instances (dropouts/non-dropouts) in the model is given by the precision parameter, which reports 0.968. The performance of the model is validated by the area under the curve parameter with a value of 0.991, and the measure to evaluate the percentage of correctly classified instances (ICC) presents a value of 96.78%.

Although the difference is minimal with other results reported in related works, it should be clarified that the research carried out did not approach the study by exclusively analyzing the academic aspect, but rather a broader set of attributes. In our case, the analysis was carried out focusing on the factors of the academic context. However, all the results report a significant incidence of academic factors in the model.

### 5.3. Testing Hypothesis 3

For this case, two attributes were validated by selecting the social factors through five selection algorithms implemented with WEKA. The selected attributes are AgeIncome_U and sex. The consideration of the social factor in comparative studies and in the evaluation of predictive models coincides with [8,12,13,30,33,43].

We can conclude that the referenced authors support the significant incidence of social factors in student dropout. Likewise, when considering these input attributes to the model (implemented with the Random Forest algorithm), the ability to correctly classify the instances (dropouts/non-dropouts) in the model is given by the precision parameter that reports 0.571, the performance of the model is validated by the area under the curve parameter with a value of 0.603, and the measure to evaluate the percentage of correctly classified instances by the ICC parameter that presents a value of 57.09%. However, [8] reports a precision of 0.887 and ICC 85.5%. Even though the difference is clearly noticeable between the evaluated parameters, it should be noted that the research

conducted by [8] does not exclusively evaluate the social aspect of the student. In our study, in the testing of the specific hypotheses, models were designed exclusively with the social factors, reporting evaluation parameters with lower values (precision = 0.571, area under the ROC curve = 0.603, and ICC = 57.09%) but equally acceptable for the model, concluding that there is a significant incidence of the student's social factors in the predictive model.

### 5.4. Testing Hypothesis 4

After making the appropriate selection of the variables that enter the predictive model, we proceeded with the training of the proposed models to choose the best algorithm that significantly detects student dropout. The evaluation parameters used in this research define Random Forest as the best classification algorithm, reporting an accuracy of 0.968. Other similar studies also defined a high accuracy for this classification algorithm [8,32,33].

### 5.5. Testing General Hypothesis

After going through a series of phases and tasks from the CRISP-DM methodology, eight classification algorithms corresponding to three techniques (decision trees, decision rules, and Bayesian networks) were analyzed. In this research, the classification technique was considered because it is the most widely used in data mining, coinciding with the results obtained by [30]. It was shown that Random Forest, which corresponds to the decision trees technique, is the best classification algorithm for the prediction of university student dropout, coinciding with the findings of [25,31]. In these works, of the 73 papers analyzed from SCOPUS and WoS sources, 67% used the decision trees technique to predict university dropout. Likewise, after analyzing 53 decision tree algorithms, Random Forest came in second place, being considered one of the best algorithms used to implement predictive models in the context of university dropout. However, other studies like [8,33,43] ranked Random Forest as the best ranking algorithm for predicting student attrition in the university ecosystem. Furthermore, [31] highlighted the use of WEKA as the most widely used data mining software tool in the studies collected, coinciding with the research of [30]. This tool was also used for the predictive purposes of dropout by [12,21,33].

In this context, it is reaffirmed that the predictive model built with the Random Forester classification algorithm, based on the input of the three factors associated with dropout, significantly detects student dropout at the National University of Moquegua with an accuracy of 0.968, a performance of 0.993 (area under the ROC curve), and a robustness of 96.78% (ICC).

### 6. Conclusions

This paper shows that training a predictive model based on DM with a dataset with attributes that characterize students from higher education studies allows for detecting student dropout at UNAM. Eight classification algorithms (Random Forest, Random Tree, J48, REPTree, JRIP, OneR, Bayes Net, and Naive Bayes) have been tested. Subsequently, performance measures were used (accuracy, ROC area, and ICC) to evaluate the models. The results show that Random Forest is the best classification algorithm to detect student dropout at UNAM, outperforming the results obtained in related works. The main limitation of the presented work is the amount of data available for each student throughout his or her university life. It would be interesting to have finer-grained data that would allow, for example, to analyze the grades obtained in each subject taken, as well as greater details of the student's social context.

For the construction of predictive models based on classification algorithms (DM techniques), an adequate balancing of the dataset was carried out to match the target class (student state) and obtain classification results with greater accuracy. For this case, the SMOTE algorithm was used. The process allowed for matching the two labels of the target class in a dataset, generating synthetic data to standardize the class into two groups (dropout/does not dropout).

The paper also analyzed the factors that influence student dropout through the study conducted. Thus, concerning the factors before university entrance, it is concluded that the attributes Type_Entry_U, YearsBetween_University_Entry, Faculty, Type_School, and Age_Entry_School have a significant impact on student desertion at UNAM. The five attributes of the pre-university entry factors have a 100% relevance in predicting the status (dropout/non-dropout) of the predictive model's target class (student status). This information is reported by the five attribute selection algorithms implemented in WEKA, namey: OneRAttributeEval, ReliefFAttributeEval, InfoGainAttributeEval, Gain-RatioAttributeEval, and SymmetricalUncertAttributeEval. In addition, the attribute quality for discriminating the target class was evaluated using the Ranker algorithm (attribute search method) combined with the five attribute selection algorithms; a weight (value) was obtained that represents the relevance of each attribute to discriminate the target class. For example, for the combination of algorithms OneRAttributeEval (attribute selection) and Ranker (search method), the results in the ranking and relevance value of each attribute were Type_Entry_U (61.102), YearsBetween_University_Entry (58.705), faculty (55.782), Type_School (52.817), and Age_Entry_School (51.114)—results that reaffirm the incidence of these attributes in the predictive model.

Concerning the factors of the student university academic context, it is concluded that the attributes SemesterIngress_U, Last_Average_Accumulated, Curricular_Plan, YearIngress_U, Curricular_Plan, Current_Subsidiary_Studies, Entry_Subsidiary, Semester_of_Tenure_U, Change_Curricular_Plan, and New_Application have a significant impact on student desertion at UNAM. According to the same five attribute selections, the ten attributes of the student university academic context factors have 100% relevance in predicting the status (dropout/non-dropout) of the target class (student status) of the predictive model algorithms. As in the previous case, the quality of the attribute to discriminate the target class was also evaluated using the Ranker algorithm (attribute search method) combined with the five attribute selection algorithms referenced above. As an example of the results, the combination of algorithms OneRAttributeEval (attribute selection) and Ranker (search method) obtained the following results for the ranking and relevance value of each attribute: SemesterIngress_U (78.995), Last_Average_Accumulated (71.678), YearIngress_U (71.215), Curricular_Plan (57.632), Current_Subsidiary_Studies (55.572), Entry_Subsidiary (54.31), Semester_of_Tenure_U (54.184), Change_Curricular_Plan (53.427), and New_Application (50.904). These results ratify the incidence of the ten attributes in the predictive model.

Finally, concerning the student social factors, it is concluded that the attributes AgeIncome_U and gender have a significant impact on student dropout at UNAM. According to the same five attribute selection algorithms, the two attributes of the student social factors have 100% relevance in predicting the status (dropout/non-dropout) of the target class (student status) of the predictive model. To assess this impact on the predictive model, the quality of the attribute to discriminate the target class was evaluated using the Ranker algorithm (attribute search method) combined with the five attribute selection algorithms referenced above. As an example of the results, the combination of algorithms OneRAttributeEval (attribute selection) and Ranker (search method) obtained the following results for the ranking and relevance value of each attribute: Age-Income_U (56.371) and gender (54.605). Again, the results confirm the incidence of the two attributes in the predictive model.

## 7. Future Work

As both current and future work, a more comprehensive comparison of predictive models using other neural networks techniques is being carried out with the dataset UNAM's DASA dataset (i.e., Logistic regression, support vector machines and nearest neighbour) in order to know the accuracy, performance, and robustness of their results for this specific problem. Furthermore, we plan to analyze the extent to which student characteristics lead to dropout (e.g., whether there is a dominant gender in dropout or whether there are degrees that are more prone to dropout).

**Author Contributions:** Conceptualization, V.F. and S.H.; methodology, V.F.; software, V.F.; validation, V.F. and S.H.; formal analysis, V.J.; investigation, V.F.; resources, V.F.; data curation, V.F.; writing—original draft preparation, V.F. and S.H.; writing—review and editing, V.F. and S.H.; visualization, S.H.; supervision, V.J.; project administration, V.J.; funding acquisition, V.F. and V.J. All authors have read and agreed to the published version of the manuscript.

**Funding:** This work was partially supported by the Spanish Government project TIN2017-89156-R, and the Valencian Government project PROMETEO/2018/002. The research was developed thanks to the support of the National University of Moquegua, which provided the information for the creation of the dataset.

**Conflicts of Interest:** The authors declare no conflict of interest.

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
