# Peer review of "Comparison of Predictive Models with Balanced Classes Using the SMOTE Method for the Forecast of Student Dropout in Higher Education"

_electronics, doi:10.3390/electronics11030457_

Round 1

Reviewer 1 Report

Reviewer's summary after reading the manuscript:

The authors argue that university student dropout is a societal issue in any country's university environment, and technological leverage is a method of developing technology ideas to address a demand that is not being addressed by current university education systems. They show that students who drop out of university studies may be identified using a predictive model trained on data mining (DM) using a dataset including features that describe students from higher education studies. The study analyzes eight predictive models to predict university dropout, which are based on data mining methods and techniques. The dataset used in the study is comprised of 4365 academic records from students at the National University of Moquegua (UNAM), in Peru. This study's goal is to establish which model provides the greatest performance indicators for forecasting and, as a result, will help to reduce student dropout. With the use of the Synthetic Minority Over-sampling Technique (SMOTE) approach for generating synthetic data, the researchers want to propose and assess the accuracy of prediction models with balanced classes. Eight classification algorithms (Random Forest, Random Tree, J48, REPTree, JRIP, OneR, Bayes Net, and Naive Bayes) were examined, with the best results coming from Random Forest. Following that, performance measurements (accuracy, ROC area, and ICC) were employed to assess the models' effectiveness. The findings demonstrate that Random Forest is the most effective classification system for detecting student drop-out at UNAM, exceeding the results reported in previous studies by a significant margin. The authors suggest that when developing prediction models based on classification algorithms, it is necessary to ensure that the dataset was properly balanced. This allows the dataset to be more closely matched to the target class (Student State) and thus provides more accurate classification results. A dataset was matched using this technique, resulting in the generation of synthetic data that was used to standardize the target class into two groups (Dropout / Does not dropout). In addition, the article looked at the variables that contribute to student dropout. The authors conclude that the prediction model based on Random Forest is the most accurate and resilient of the models tested.

----------------------------------------

Dear authors, thank you for your manuscript. I enjoyed reading it. Presented are some suggestions to improve it:

(1) Please consider modifying the title of the manuscript to include the words "using the SMOTE method" so that it would be easier for potential readers to find your study. Please kindly elaborate on what the "SMOTE" acronym means early in the manuscript. It was not explained in the text until page 4 although the SMOTE acronym had already been mentioned several times on pages 1-3 prior to that.

(2) Please include a "Limitations" section to discuss what were the challenges faced, and how your team overcame those challenges. This would be very beneficial to the readers as they would be able to learn from your expert knowledge.

(3) To improve the impact and readership of your manuscript, the authors need to clearly articulate in the Abstract and in the Introduction sections about the uniqueness or novelty of this article, and why or how it is different from other similar articles. Can the authors please kindly elaborate more about how this study is relevant to "electronics" since it was submitted for publication in the journal entitled "Electronics"?

(4) Please substantially expand your review work, and cite more of the journal papers published by MDPI.

(5) All of the references cited are not yet properly formatted according to MDPI's guidelines. For example, the DOIs of all the journal papers cited are not included yet. For the references, instead of formatting "by-hand", please kindly consider using the free Zotero software (https://www.zotero.org/), and select "Multidisciplinary Digital Publishing Institute" as the citation format, since there are currently 44 citations in your manuscript, and there may probably be more once you have revised the manuscript.

Thank you.

Author Response

First of all, we would like to thank the reviewer for their constructive remarks on the previous version of this paper. We have incorporated all suggested changes in this new version. We believe that the presented version has adequately addressed all the comments, with appropriate explanations.

(1) Please consider modifying the title of the manuscript to include the words "using the SMOTE method" so that it would be easier for potential readers to find your study. Please kindly elaborate on what the "SMOTE" acronym means early in the manuscript. It was not explained in the text until page 4 although the SMOTE acronym had already been mentioned several times on pages 1-3 prior to that.

Reply: According to the reviewer’s suggestion, we have changed the title of the paper

(2) Please include a "Limitations" section to discuss what were the challenges faced, and how your team overcame those challenges. This would be very beneficial to the readers as they would be able to learn from your expert knowledge.

Reply: According to the reviewer’s suggestion, we have included the main limitations of the proposed study. Specifically, conclusions have been extended with the identified limitations. 

(3) To improve the impact and readership of your manuscript, the authors need to clearly articulate in the Abstract and in the Introduction sections about the uniqueness or novelty of this article, and why or how it is different from other similar articles. Can the authors please kindly elaborate more about how this study is relevant to "electronics" since it was submitted for publication in the journal entitled "Electronics"?

Reply: According to the reviewer’s suggestion, the abstract and intro have been modified, including the importance of the proposed study and also highlighting the improvement in results obtained compared to other proposals. 

(4) Please substantially expand your review work, and cite more of the journal papers published by MDPI.

Reply: According to the reviewer’s suggestion, more MDPI papers have been included in the paper.

(5) All of the references cited are not yet properly formatted according to MDPI's guidelines. For example, the DOIs of all the journal papers cited are not included yet. For the references, instead of formatting "by-hand", please kindly consider using the free Zotero software (https://www.zotero.org/), and select "Multidisciplinary Digital Publishing Institute" as the citation format, since there are currently 44 citations in your manuscript, and there may probably be more once you have revised the manuscript.

Reply: According to the reviewer’s suggestion, we have adapted all the paper references.

Reviewer 2 Report

The article presents a comparison of predictive models with balanced classes for forecasting student dropout in higher education. In my opinion, the topic is very interesting, as part of this article, the authors analyzed eight predictive models for prediction of university dropout, based on data mining methods and techniques, using WEKA to implement it, with a data set of 4365 academic records of students from the National University of Moquegua (UNAM).

The author showed that it is possible to detect the resignation of students at this university. One of the tested algorithms (Random Forest) turned out to be better than the other algorithms. This article is well-worded and has good consistency logic. The article presents a series of tests that are intended to enable the reader to understand the proposed method in this article. However, I have remarks for the authors:

 1. It seems to me that if the number of records were greater, the results would be more reliable.

2. Equation number (1) is not well described.

3. Minor editorial mistakes that need to be corrected.

Author Response

First of all, we would like to thank the reviewer for their constructive remarks on the previous version of this paper. We have incorporated all suggested changes in this new version. We believe that the presented version has adequately addressed all the comments, with appropriate explanations.

  1. It seems to me that if the number of records were greater, the results would be more reliable.

Reply: Yes, we agree with the reviewer, with a larger number of data, probably better results would be obtained. The model uses a dataset of 4,365 student academic records, from 2008 to 2019, unfortunately, it is not possible to obtain a larger dataset.

  1. Equation number (1) is not well described.

Reply: According to the reviewer’s suggestion, we have changed the equation, there was an error in it

  1. Minor editorial mistakes that need to be corrected.

Reply: The paper has been reviewed. Some typos have been corrected and many sentences have been modified.

Reviewer 3 Report

The paper is nicely written. I have the following comments:

  1. In the related work section, highlight the limitations of related work and show how you will address these limitations
  2. The main contributions of the work is not clear.
  3. Figure 1 is unreadable and very small. Please fix this issue
  4. The font size in figure 2 is very small and unreadable. Please fix this issue. You may divide this figure into 2 figures
  5. Model design section is missing. For example, what are the parameters of each model? input / output?
  6. In Table 1, the Target Class StatusStudent is numeric. Is this correct? if so, the problem is regression and not classification. Please comment on this

Author Response

First of all, we would like to thank the reviewer for his/her constructive remarks on the previous version of this paper. We have incorporated all suggested changes in this new version. We believe that the presented version has adequately addressed all the comments, with appropriate explanations.

1. In the related work section, highlight the limitations of related work and show how you will address these limitations

Reply: According to the reviewer’s suggestion, we have included the main limitations of the proposed study. However, we have thought of putting it in the conclusions section. Thus, conclusions have been extended with the identified limitations.

2. The main contributions of the work is not clear.

Reply: According to the reviewer’s suggestion, the abstract and intro have been modified, including the importance of the proposed study and also highlighting the improvement in results obtained compared to other proposals.

3. Figure 1 is unreadable and very small. Please fix this issue

Reply: According to the reviewer’s suggestion, Figure 1 has been extended.

4. The font size in figure 2 is very small and unreadable. Please fix this issue. You may divide this figure into 2 figures

Reply: According to the reviewer’s suggestion, Figure 2 has been divided in two, allowing a better view of the texts that appear in the figure.

5. Model design section is missing. For example, what are the parameters of each model? input / output?

Reply: According to the reviewer’s suggestion, the modeling phase has been extended, including more information about the experiments. Moreover, a new table (Table 2), including the used parameters, was created.

6. In Table 1, the Target Class StatusStudent is numeric. Is this correct? if so, the problem is regression and not classification. Please comment on this

Reply: There was an error in the data type of this attribute; the attribute has only two values (dropout, not dropout); therefore, what is done is to classify one of these two possible values.